

# First magmatism in the New England Orogen, Australia: Forearc and arc-backarc components in the Bakers Creek Suite gabbros

Seann J. McKibbin[1,2,*], Bill Landenberger[1], Mark C. Fanning[2]

[1]School of Environmental and Life Sciences, University of Newcastle, University Drive, Callaghan, 2308, Australia
[2]Research School of Earth Sciences, Australian National University, Bldg. 61, Mills Road, Canberra, 0200, Australia
*now at: Analytical, Environmental and Geo-Chemistry, Vrije Universiteit Brussel, Pleinlaan 2, Brussels 1050, Belgium

*Correspondence to*: Seann J. McKibbin (seann.mckibbin@gmail.com; seann.mckibbin@vub.ac.be)

**Abstract.** The New England Orogen, Eastern Australia, was established as an outboard extension of the Lachlan Orogen through the migration of magmatism into the forearc basin and accretionary prism. Widespread S-type granitic rocks of the
Hillgrove and Bundarra Supersuites represent the first pulse of magmatism, followed by I- and A-types typical of circum-Pacific extensional accretionary orogens. Associated with the former are a number of small tholeiite-gabbroic to intermediate bodies of the Bakers Creek Suite, which are a heat source for production of granitic magmas and potential tectonic markers indicating why magmatism moved into the forearc and accretionary complexes, rather than rifting the old Lachlan Orogen arc. The Bakers Creek suite gabbros capture an early (~305 Ma) forearc basalt-like component with low Th/Nb and with
high Y/Zr and Ba/La, recording melting in the mantle wedge with little involvement of a slab flux and indicating forearc rifting. Subsequently, arc-backarc like gabbroic magmas (305-304 Ma) were emplaced followed by diverse magmatism of mixed compositions leading up to the main S-type granitic intrusion (~290 Ma). This trend in magmatic evolution implicates forearc and other mantle wedge melts in the heating and melting of fertile accretion complex sediments and relatively long (~10 Myr) timescales for such melting.

# 1 Introduction

The New England Orogen (NEO) is the youngest and easternmost component in the Tasmanides accretionary orogenic system and of the Australian continental craton (e.g. Cawood 2011; Figure 1a). The NEO has similarities to its older neighbour the Lachlan Orogen, such as west-dipping subduction (e.g. Leitch 1974; 1975), a general tectonic regime switching between crustal thinning and thickening (Collins 2002; Brown 2003), and granitic magmatism spanning a
compositional range between peraluminous and metaluminous end members (S-type and I-type for igneous and sedimentary sources respectively: Hensel et al. 1985; Chappell and White 2001; Collins and Richards 2008). However, the NEO represents outboard migration of magmatic activity into the Lachlan Orogen forearc basin and accretionary prism sediments (Jenkins et al. 2002). These sediments, derived from the 'calc-alkaline' arc rocks, inherited juvenile isotopic characters that were passed on to their derivative granitic melts by rapid subduction cycling (Kemp et al. 2009).



The termination of the Devonian-Carboniferous magmatic arc and replacement by widespread and relatively disorganised magmatism (Collins et al. 1993; Caprarelli and Leitch 1998; Jenkins et al. 2002) at ~290 Ma, culminated in the emplacement of the contrasting Bundarra and Hillgrove S-type granitoid supersuites (Rosenbaum et al. 2012). They differ from each other with the former being a voluminous, compositionally homogenous belt, while the latter is variably foliated, generally more

mafic in composition (Shaw and Flood 1981), and is associated with high-temperature low-pressure (HTLP) metamorphic complexes (Farrell 1988; Dirks et al. 1992) and small, mafic to intermediate intrusive bodies referred to as Bakers Creek Suite (Jenkins et al. 2002). Following minor magmatic activity (Roberts et al. 1995; Donchak et al. 2007; Cross et al. 2009; Phillips et al. 2011) and a temporal magmatic gap associated with orogeny (Rosenbaum et al. 2012), the NEO was overwhelmed by voluminous I-type magmatism from ~265 Ma (Li et al. 2012).

The mafic mantle-derived plutons of the Bakers Creek Suite, while small and variably evolved, ultimately records the conditions of mantle partial melting and subduction zone contributions to the first magmatism in the NEO. New advances in the understanding of the geochemistry of arc-related magmas have established roles for the various mafic magmas emplaced during subduction zone initiation and migration. These include basalts with forearc (FAB; Reagan et al. 2010; Meffre et al.

2012; Ribeiro et al. 2013), backarc (BAB; Langmuir et al. 2006; Pearce and Stern 2006), and early arc tholeiite (EAT; Todd et al. 2012) affinities. Each of these has distinctive trace element compositions which can potentially be recognised in paleo-arc systems (Dilek and Furnes 2014, Pearce 2014). We present here a study of the geochemistry of the Bakers Creek Suite with emphasis on samples from uncontaminated, mafic plutons, and U-Pb chronology of these earliest magmatic rocks in the NEO. Further, we identify forearc and backarc components, and address the tectonic setting and mechanisms by which

magmatism began in this section of an ancient extensional accretionary orogen.

## 2 Regional Geology

The Southern NEO is built upon a metasedimentary base comprising the Tamworth Belt (representing an old forearc basin) and the Tablelands Complex (an accretionary prism) separated by the Peel-Manning Fault System (Leitch 1974; Korsch 1977). Both are related to Devonian-Late Carboniferous magmatic arc rocks (Leitch 1975). In the Tablelands Complex (Fig.

1b), high temperature and low pressure metamorphism overprints the accretion-subduction sequences (Wongwibinda and Tia Complexes: Farrell 1988; Hand 1988; Dirks et al. 1992; Phillips et al. 2008; Craven et al. 2012). Subsequently, intrusion of the Hillgrove Suite biotite granites and granodiorites (±garnet, hornblende) took place, forming a discontinuous belt of scattered plutons (Flood and Shaw 1977; Shaw and Flood 1981). Spatially associated with the Hillgrove granitoids are the small plutons of the Bakers Creek Suite, a diverse group of mafic to intermediate bodies ranging from two-pyroxene

(±olivine) gabbros and related cumulate rocks through hornblende-biotite diorites to mafic hornblende-bearing granodiorites (Jenkins et al. 2002). The Hillgrove and Bakers Creek mafic plutons have been exhumed as a result of early Permian rifting and subsequent thrusting during the Hunter-Bowen Orogeny (Figure 1b; Landenberger et al. 1995; Shaanan et al. 2015).



Also present are the voluminous and more strongly peraluminous S-type granites of the Bundarra Suite, lying in a continuous north-trending belt to the west of the Hillgrove Suite (Flood and Shaw 1977; Shaw and Flood 1981). In contrast to the Hillgrove Suite, the Bundarra Suite granites are generally non-foliated, have no mafic plutons associated with them, and are not associated with metamorphic complexes, despite generally contemporaneous intrusion (Rosenbaum et al. 2012).

Mafic, primitive members of the Bakers Creek Suite include the small Barney House and Big Bull gabbros, while larger plutons such as the Days Creek gabbro and Apsley River Complex exhibit more complex characteristics of differentiation (e.g. samples BHC2, CC26A, G39, and GK2 respectively from Jenkins et al. 2002). Sampling was undertaken with a focus on mafic plutons such as the Barney House, Big Bull, and Days Creek gabbros. The Barney House and Big Bull gabbros are
small (scale of tens to hundreds of metres) and consist of finely crystalline gabbro, often hosting plagioclase phenocrysts, in contact with low-grade metasedimentary country rock. The Big Bull gabbro occurs as the most mafic member in a full spectrum of rocks varying from mafic to felsic (Sheep Station Creek Complex). In contrast, the Days Creek gabbro occurs as two larger plutons (~1 and ~2 km in length) partially surrounded by the Tobermory monzogranite (Hillgrove Suite) except at the southern margin where it borders turbidites of the Girrakool Beds. It is dominated by medium to coarse grained gabbro
and contains rare pegmatite (grainsize 5-20 mm; Figure 2a). The southern pluton exhibits a doleritic (~1 mm) chilled margin against turbidites which are contact-metamorphosed and exhibit rare occurrences of melting. Widespread but poorly exposed pieces of dolerite are found at various locations across both plutons, some in association with metasedimentary rocks and felsic veins containing gabbro breccia (Figure 2b). The encompassing Tobermory monzogranite is usually coarse (average grainsize a few mm) and lacks foliation. Although most contacts are not exposed, it is often finer grained (to ~1 mm) nearer
the gabbro, indicative of quenching and late emplacement relative to most other members of the Hillgrove Suite (Landenberger et al. 1995). The Tobermory monzogranite is cut on the western side by younger, unrelated mid-late Permian to Triassic I-type granite (Li et al. 2012).

## 3 Analytical Methods

Selected 30 μm thin sections of samples were polished and carbon coated for X-ray analysis of mineral phases by scanning
electron microprobe (SEM) at the University of Newcastle (UoN) using a Phillips XL30 SEM, with Oxford ISIS Energy Dispersive Spectrometer (EDS); 15 kV accelerating voltage; 3 nA beam current. Bulk-rock samples were crushed by tungsten-carbide mill and diluted in lithium borate flux at 1050 °C to produce a glass disc. Major oxide and some trace elements were analysed by X-ray fluorescence spectrometry (XRF) at the UoN (Spectro X'Lab 2000 XRF system with: EDS; Pb anode tube; polarised beam; multiple targets). All Fe is reported as FeO. Glass discs from this study and from
Jenkins et al. (2002) were sectioned and polished for trace element analysis by laser ablation inductively coupled plasma mass spectrometry at the Research School of Earth Sciences, Australian National University (ANU), using a quadrupole Agilent 7500s coupled to a 193 nm ArF Excimer laser (Eggins 2003). Samples were run against NIST glasses and either [43]Ca



or $^{29}$Si were used as internal standards depending on bulk silica content (using CaO or SiO$_2$ from XRF) and data reduced using an in-house spreadsheet.

Magmatic zircon $^{238}$U/$^{206}$Pb ages of gabbroic and dioritic samples were determined at the ANU using Sensitive High-mass

Resolution Ion Micro Probes (SHRIMP). The gabbroic samples (Barney House and Days Creek Gabbro) were analysed using SHRIMP-RG against the reference standard TEMORA, while dioritic samples (Bakers Creek Complex and Charon Creek Diorite) were analysed using SHRIMP-I against the AS3 reference material. Rejection of analyses was made on the basis of measureable common Pb, loss of Pb, or contribution to an unreasonably high MSWD. In reviewing other U-Pb data for the NEO in the literature, it is noted that they were obtained against a range of reference materials over many years. The

comparative study of Black et al., (2003) showed that some zircon ion-probe reference materials yielded small biases, with ages calculated against AS3 being ~1% too high, and ages calculated against SL13 being variably (although on average, ~1%) too low. To account for this, we made corrections of -1% to our AS3 ages and +1% to SL13 ages assembled in our age compilation; other relevant AS3 ages in the literature were verified by other standards (Roberts et al. 2004; 2006). Although these corrections are significant in terms of precision, they have little influence on tectonic conclusions.

More importantly, some of the U-Pb ages for early NEO magmatism, the S-type Rockvale Granodiorite and Tia Granodiorite, as well as the I-type Halls Peak Volcanics, appear biased towards younger ages by rejection criteria. Cawood et al. (2011) presented ages for these and other igneous bodies, undertaken against reliable standards (CS3) that do not require corrections of the kind discussed above. However, they included an arbitrary criterion for recognition of zircon

inheritance, that analyses older than 300 Ma should be excluded. Because individual zircon U-Pb determinations for these samples have approximately Gaussian distributions centred near 300 Ma, this has led to an excessive number of rejections, and naturally to ages <300 Ma (Rockvale Granodiorite: 292.6 ±2.4 Ma, MSWD 1.5, 10 from 30 rejected; Tia Granodiorite: 295.7 ±2.8 Ma, MSWD 0.37, 14 from 27 rejected; and Halls Peak Volcanics: 292.6 ±2.0 Ma, MSWD 0.68; 11 from 26 rejected). An alternative criterion for recognition of zircon inheritance follows from the observation of Jeon et al., (2012)

that the Th/U ratios of obviously inherited zircon in the Bundarra Supersuite are generally greater than ~0.3, while new magmatic zircon extends to as low as ~0.05. We have recalculated the ages of these samples with an emphasis on including zircon with low Th/U and maintaining Gaussian distributions.

## 4 Petrography

Fine grained, doleritic Barney House, Big Bull (Sheep Station Creek Complex) and Days Creek gabbro exhibit granular and

flow foliated (Figure 3a) to ophitic-subophitic textures (Figure 3b), and occasionally contain phenocrystic or glomerocrystic plagioclase (Figure 3c). Plagioclase is elongate, sub-rectangular or lath-like, sharing irregular edges with or enclosing olivines, and is typically normally zoned or unzoned, but sometimes contains distinct cores (An$_{72-60}$ in groundmass; mostly



~$An_{80}$ but up to $An_{86}$ in cores). Phenocryst rims are sodic (to $An_{50}$) and texturally interlock with the fine gabbro groundmass. Olivine ($Fo_{62-72}$) is common and exhibits rounded, irregular, and embayed morphologies or very rarely interstitial and mantled by pyroxene or hornblende. Pyroxene occurs as oikocrysts and interstitial crystals. High-Ca clinopyroxene (diopside-augite Mg# 78) is more common than low-Ca orthopyroxene (Mg# 71) and contains exsolution lamellae of the latter. Pyroxene shares interstices with calcic magnesio-hastingsite hornblende and pargasite hornblende (Mg# 71) with high $TiO_2$ (~3.2 wt. %) and $Al_2O_3$ (~11.0 wt. %), as well as small titanium-rich phlogopite (~3.9 wt. % $TiO_2$; average Mg# 75). Ilmenite and rare magnetite (sometimes intergrown) is usually associated with amphibole and phlogopite, often being mantled by them or included in their interstitial domains.

Coarsely crystalline gabbro (mm to cm crystals) is more typical of the Days Creek gabbro, with orthocumulate or mesocumulate textures (Figure 3d) comprising plagioclase, rare resorbed olivine, high-Ca clinopyroxene, very rare low-Ca orthopyroxene, ilmenite and amphibole (latter often secondary). Massive coarse grained gabbro also has rare granular texture (Figure 3e). Plagioclase ($An_{80-47}$) ranges from isolated, equant euhedral crystals to subhedral crystals in an interlocking network, defining the ortho- or mesocumulate texture. They are commonly normally zoned, and rarely exhibit oscillatory zoning or scissor deformation twins. Olivine is $Fo_{65-59}$, is anhedral or embayed. Secondary clinozoisite and serpentinite after plagioclase and olivine was not observed in fine grained gabbros but is present in coarse grained samples. Pyroxenes are subhedral or interstitial and rarely optically continuous across multiple domains. In coarse gabbros, high-Ca clinopyroxene is diopside-augite (average Mg# 79) while low-Ca orthopyroxene is very rare, possibly because of uralitisation (Mg# 70 with exolved clinopyroxene at Mg# 78). Very fine orthopyroxene exsolution is also present in clinopyroxenes. Amphibole is present in abundances approximately equal to that of pyroxenes and occurs as primary interstitial magnesio-hornblendes (pale brown and green; average Mg# 63) and secondary fibrous or radiating irregular actinolitic-hornblende, magnesio-hornblende or tschermakitic-hornblende (green to green-blue varieties; average Mg# 60). Anhedral or interstitial ilmenite shares intercumulus spaces with pyroxenes and amphiboles.

Bakers Creek Suite rocks with higher silica contents (geochemically intermediate between gabbros and granitoids of the Hillgrove Suite) display a wide range of textures and variation in mineralogy. Poikilitic, equigranular and foliated textures are observed in rocks from parts of the Days Creek Gabbro, Camperdown Complex and Woodburn Diorite. In poikilitic biotite-diorite associated with the Days Creek gabbro, small rectangular or subhedral plagioclase and granular orthopyroxene is randomly enclosed within large oikocrysts of biotite, quartz and orthopyroxene (sample DC98; Figure 3f). Equigranular diorite from the Camperdown Complex (CC11) is dominated by subhedral plagioclase with green amphibole in large interstitial quartz domains, with possibly secondary green amphibole and calcite. More felsic varieties of the Bakers Creek Suite are closer in composition to and continuous with that of the Hillgrove Suite granitoids, and have developed tectonic foliations or sub-gneissic textures: e.g. the Woodburn diorite is composed of subhedral plagioclase and amphibole, folded or





kinked biotites and interstitial or ophitic quartz domains (WB32). Quartz and plagioclase are occasionally graphically intergrown. Preferentially aligned biotite is the main contributor to foliation

# 5 Geochemistry

Samples of the Bakers Creek Suite cover a broad geochemical range (Figure 4a and Supplement 1), with most being gabbroic (48-52 wt. % $SiO_2$), one more mafic (~45.5 wt. % $SiO_2$), and some dioritic (basaltic-andesitic or andesitic compositions up to 61.3 wt. % $SiO_2$). We also report one granitic sample from the Eastlake monzogranite, associated with the Woodburn diorite (~70 % $SiO_2$). For the finely crystalline gabbroic or doleritic samples, FeO, $Na_2O$, $P_2O_5$ and $TiO_2$ generally increase as MgO, CaO and $Al_2O_3$ decrease (Figure 4b); they define a trend that passes through the Global mid-ocean ridge basalt (MORB) composition for major elements (Arevalo and McDonough 2010). The Big Bull gabbro (CC26A) is a close but resolvable outlier to this trend for some elements (e.g. FeO). From their petrography they likely represent liquid compositions; a range of FeO/MgO for similar $SiO_2$ is therefore indicative of tholeiitic style evolution (Arculus 2003; Zimmer et al. 2010). In contrast, coarsely crystalline gabbros do not follow this trend for most oxides. With decreasing MgO, they have increasing $Al_2O_3$, $P_2O_5$ and $TiO_2$ but FeO, CaO, $Na_2O$, $SiO_2$ and $K_2O$ are poorly related to MgO (e.g. Figure 4b). All gabbroic samples, whether coarse or finely crystalline, have low $K_2O$ (<0.3 wt. %; low-K association of Gill 1981) and are 'basaltic' (Le Bas and Streckeisen 1991), with dioritic samples spanning the range between Bakers Creek gabbroic and Hillgrove Suite granitic compositions through 'basaltic andesite' to 'dacite/rhyolite' (e.g. ~52-62 wt. % $SiO_2$ and 0.5-2.0 wt. % $K_2O$) (Figure 4a). Some samples deviate from this trend to higher or lower levels of minor element contents (e.g. MP2, CCD, and GK5, FHB respectively). The granitic sample (EA31) falls within or near the main group of Hillgrove Suite samples for all major elements (e.g. Shaw and Flood 1981).

There is a general correlation between major element and trace element geochemistry, with gabbroic samples having MORB-like trace and rare earth element (REE) abundances, while samples with intermediate/granitic geochemistry are enriched in incompatible trace and REEs (Figure 5; multi-element plots normalised to Global MORB of Arevalo and McDonough 2010). In detail, most of the finely crystalline gabbros display flat and MORB-like trace element patterns, especially for high-field strength elements (HFSE) Zr, Hf, Ti, Y as well as REE, but with positive Cs, Rb, Ba, Th, U and Pb anomalies and depletions in Nb and Ta (e.g. Barney House and Big Bull gabbros). Some coarsely crystalline gabbros (DC15, DC16) are geochemically similar to the fine grained gabbros, but most are variably depleted in HFSE, REE, Th, Nb, Ta, and P, with positive anomalies for the same elements as in the finely crystalline gabbros (Cs, Rb, Ba, Th, U, and Pb) but also including Eu. Cr and Ni are variable, with some exhibiting clear enrichments (e.g. GK5 and FHB). Coarsely crystalline samples from the Days Creek Gabbro with middle and high-range Mg# display erratic, concave up patterns, with elevated LILE and Sr, low Nb, Ta and HFSE and higher abundances of Cr and Ni. REE patterns are flat with considerable variation in absolute abundances and, as for finely crystalline samples this is generally correlated with FeO. For samples with higher





Mg#, REE abundances are lower and distinct Eu peaks and light REE depletions are apparent. Some gabbros from the Days Creek Gabbro appear to be anomalous (D12, DC104, and DC65) with variable depletion in Cs, Th, U, Nb, Ta, K, P, Zr, Hf, and light REE, and potential enrichment in Ba.

Trace and REE concentrations for higher silica, geochemically 'intermediate' rocks of the Bakers Creek Suite, as well as the Eastlake monzogranite sample (Hillgrove Suite; EA31) are variably higher and patterns are inclined; peaks in Cs, Th, U, K, Pb, Zr and Hf alternate with negative Nb, Ta, Sr, P and Ti anomalies. The Hillgrove Suite sample is the most enriched in incompatible trace elements, and other geochemically 'intermediate' samples also exhibit generally intermediate concentrations of such elements, i.e. they are correlated with $SiO_2$. Negative anomalies are common for Ba, Nb and Ta, with

some variation in HFSE where concentrations are similar to the granites. Cr and Ni likewise display a range intermediate to the gabbros and granites. Though higher silica samples of the Bakers Creek Suite have compositions intermediate to the gabbros and the monzogranite for most elements, there are important exceptions for Sr, P, Ti, Eu and heavier REEs for certain samples (CCD, MP2, DC98).

## 6 Zircon Chronology

We find zircon $^{206}Pb/^{238}U$ ages of 303.9 ±3.2 Ma (15 points with no rejections, MSWD 1.7) for the Barney House Gabbro and 305.1 ±2.9 Ma (18 points with 1 rejection, MSWD 1.5) for the Days Creek Gabbro. These are the oldest ages for intrusive rocks in the Southern NEO, with the possible exceptions of the Rockvale and Tia Granodiorites (see below). The Bakers Creek Complex has a similar age of 299.3 ±3.1 Ma (corrected for AS3; uncorrected age is 302.3 ±3.1 Ma, 18 points with 2 rejections, MSWD 1.3), while the Charon Creek Diorite has a younger age of 290.4 ±3.2 Ma (corrected for AS3;

uncorrected age is 293.3 ±3.2 Ma, 15 points and 5 rejections on the basis of Pb-loss or measureable common Pb, MSWD 1.6). U-Pb data are given in Figure 6 and Supplement 2.

With our Th/U criteria, the recalculated age of the Rockvale Granodiorite is 296.7 ±2.3 Ma (MSWD 1.8), ~4 Myr older than the age given by Cawood et al. (2011) (Supplement 3). We revisited the original U-Pb age for the Rockvale Granodiorite

reported by Kent (1994), which at 303 ±3 Ma, is older than other igneous rocks of the Southern NEO. His rejection criteria were fundamentally in accord with ours, although the age may suffer from variable bias from the SL13 standard which would have depressed the age. If bias was, in this case, insignificant (SL13 behaviour is not consistent and sometimes does not bias ages at all; Black et al., 2003) then a discrepancy of 1.0 Myr remains between Kent (1994) and the age recalculated from the data of Cawood et al. (2011). If, however, bias is present then the age could be up to ~306 Ma, with associated

discrepancy of up to ~4 Myr. Hence, U-Pb data for the Rockvale Granodiorite remains poorly understood.



Our recalculated age of the Tia Granodiorite is 299.7 ±2.0 Ma (MSWD 0.92), again ~4 Myr older than reported by Cawood et al., (2011). This age is consistent with previous U-Pb determinations (~300 and ~302 Ma from Dirks et al. 1993 and Kemp et al. 2009) which, together with ages for the Rockvale Granodiorite, imply that the intrusion of some Hillgrove Supersuite plutons considerably predated the main S-type flux represented by the Bundarra Supersuite and most of the Hillgrove Supersuite. For the Halls Peak Volcanics, the recalculated age is 295.7 ±2.2 Ma (MSWD 1.6), ~3 Myr older than given by Cawood et al., (2011). Selected and rejected zircon analyses from Cawood et al. (2011) are given in Supplement 3.

## 7 Discussion

### 7.1 Tectonic Setting: BAB, EAT, FAB, or something else?

Because a wide range of compositions are present in chilled margins, including anomalous (e.g. DC65) and main-group samples (e.g. BH30) of various MgO contents, magmatic differentiation occurred before or during emplacement of magmas at depth in the mantle wedge, or during ascension through the mantle wedge and overlying crust. Some differentiation also occurred in situ, indicated by coarsely crystalline samples DC15 and DC16 which are geochemically similar to quenched or finely crystalline gabbros.

Melt compositions lie in the MORB and slab-distal BAB and FAB fields in terms of Ti and V (Shervais 1982; Pearce 2014; Figure 7a) and seem to have been only subtly affected by subduction zone influences on these elements (McCulloch and Gamble 1991; Woodhead et al. 2001). Alternately, Y/Zr can be used to identify previously depleted mantle sources (e.g. Arculus et al. 2015). Despite the geochemical similarity of these elements to V/Ti under typical subduction zone redox conditions (trivalent and tetravalent respectively) there is clear distinction between main-group Bakers Creek Suite melt compositions and anomalous melts in Y/Zr space (Figure 7b; samples DC65, DC104 and D12). These have much lower Zr (and Hf) for similar Y contents, which is a characteristic shared by forearc type basalts, e.g. Izu-Bonin FAB (Reagan et al. 2010; Ribeiro et al. 2013; Arculus et al. 2015).

Despite MORB-like Ti-V and Zr-Y systematics for most Bakers Creek Suite melt compositions, a subduction-derived component is clear for 'slab flux' elements. In Nb/Yb and Th/Yb space (Pearce 2014) the main group of samples are well clear of the MORB-OIB array, and are high in the 'Oceanic arcs' region (Figure 8a), while two anomalous samples again share similarities with FAB-type basalts in having very low Th/Yb, consistent with Zr-Y (but not Ti-V) systematics. Multi-element plots in Figure 5 suggests this is due to a combination of (1) Nb depletion, either by retention in the mantle-wedge source or an under-contribution from the slab; and (2) addition of Th (and U) to main group Bakers Creek Suite melts via addition of a sedimentary component (e.g. Woodhead et al. 2001). The latter might have been derived from subducted sediments in the undergoing slab, or by simple contamination with Lachlan Orogen accretionary prism material as metasediments or S-type granitic melts (i.e. Hillgrove Suite). Anomalous Bakers Creek Suite melts DC65 and DC104 are in



the extension of the MORB array to very low Th/Yb and low or very low Nb/Yb. Along with Th, the similarly incompatible indicators of sedimentary melting U and LREE (especially La and Ce) seem to have been under-contributed (no melting of zoisite or allanite; Spandler and Pirard 2013), although the Th/U ratio of DC65 and DC104 is much lower than other samples. DC65 and D12 received unusually high Ba contributions, which may indicate a distinct fluid component

(Woodhead et al. 2001). Anomalous samples are therefore associated with a peculiar elevated Ba/La (Figure 8b). Additionally, DC104 has much lower Cs (and $K_2O$) than the others, despite similar levels of Rb in main-group and anomalous Bakers Creek Suite samples. This strongly indicates decoupling of Rb from other trace alkalis Cs and $K_2O$, as well as from Ba, elements that are ordinarily associated in sub-arc settings, e.g. via phengite and paragonite melting (Spandler and Pirard 2013).

A direct comparison is made of the multi-element plots for the melt compositions of the Bakers Creek Suite, with the FAB basalts of Reagan et al. (2010) and Ribeiro et al. (2013) in Figure 9, for similar major element compositions (especially for MgO, in the range ~6.6-8.6 wt. %). They share some relative and absolute abundance trace element characteristics, especially those of anomalous composition (D12, DC104, and DC65). Low abundances of certain slab-flux elements such as

Th and U, the light to mid-REEs especially La and Ce (and consequently low LREE/HREE), and the HFSE Zr and Hf, all indicate involvement of a FAB component in otherwise arc- or backarc-like basaltic compositions.

The trace element geochemistry of Bakers Creek Suite samples therefore indicates divorcement of some components ordinarily associated in subduction zone associations (e.g. Reagan et al. 2010; Ribeiro et al. 2013). The unusual

compositions found in the chilled margin of the Days Creek Gabbro may represent early forearc style magmas, with the more extreme characteristics, especially high Y/Zr and low Th/Yb. As chilled margins, these have been specifically sampled in the field and might be less often captured by random undersea sampling (e.g. Arculus et al. 2015). Such magmas were overwhelmed by later arc-backarc style magmas (main group Bakers Creek) and the larger Days Creek gabbro might be an example of a feeder pipe, capturing early and late components.

**7.2 Chronology of earliest NEO magmatism**

The oldest dated Southern NEO intrusives clearly comprise the gabbroic plutons of the Bakers Creek Supersuite (Barney House and Days Creek Gabbro); they are clearly resolved by U-Pb dating from other intrusive bodies. Other early samples which are not so clearly resolved include the largest compositionally-intermediate pluton of the Bakers Creek Supersuite (Bakers Creek Complex), as well as isolated members of the S-type Hillgrove Supersuite (Tia and Rockvale Granodiorites,

with the age of the latter not well known but still older than 294 Ma; possibly also the Blue Knobby Monzogranite and Henry River Granite). The Tia Granodiorite constrains the age of the HTLP Tia Complex (Phillips et al., 2008) to greater than 299.7 ±2.0 Ma (recalculated here from data of Cawood et al. 2011); metamorphic zircon in the HTLP Wongwibinda Complex records a similar, or slightly younger, U-Pb age of 296.8 ±1.5 Ma (Craven et al. 2012).





As the magmatic pulse accelerated, diverse compositions continued with intrusion of the Jibbinbar Granite at ~298 Ma (Cross et al. 2009) followed by the Rockisle Granite, Dorrigo Mountain Complex, and Mount You You Granite at ~295 Ma (Rosenbaum et al., 2012). This diverse magmatism is also reflected at the same time in volcanic rocks, with the I-type Halls Peak Volcanics and various basaltic flows near the base of the newly opened Barnard Basin (Cawood et al. 2011), and the Alum Rock Volcanics (Roberts et al. 1996).

The major phase of pre-Hunter-Bowen magmatism in the Southern NEO occurred with the climactic emplacement of S-type granites and granodiorites of the Bundarra Supersuite at ~292-285 Ma, and many larger plutons of the Hillgrove Supersuite at ~293-288 Ma (e.g. Hillgrove Monzogranite). Some diversity in magmatic compositions continued throughout this period, with emplacement of the ungrouped Kaloe Granodiorite (Cawood et al. 2011), Bullaganang Granite (Donchak et al. 2007) and Gandar Granodiorite (Rosenbaum et al. 2012), as well as our own AS3-corrected age for the Charon Creek Diorite (Bakers Creek Suite).

S-type magmatism persisted until ~280 Ma for the Cheyenne Complex of the Hillgrove Supersuite, and possibly until ~282 Ma for part of the Banalasta Monzogranite of the Bundarra Supersuite (Phillips et al., 2011). This last emplacement of S-type magma was contemporaneous with another burst of I-type magmatism in the form of the Alum Mountain Volcanics (~274 Ma; Roberts et al. 1995) and the more conspicuous low-K, HREE-depleted Greymare Granodiorite (Donchak et al. 2007) similar to the Clarence River Supersuite, and finally the ~267 Ma Barrington Tops Granodiorite (Cawood et al. 2011). This chronology is illustrated in Figure 10 (see also Supplement 4); thereafter, magmatism following the Hunter-Bowen Orogeny is reviewed by Li et al. (2012).

## 7.3 Tectonic Implications

Magmatism in the long lived subduction zone of the Lachlan Orogen ceased at ~305 Ma (Claoué-Long and Korsch 2003; Roberts et al. 2004; 2006; Jeon et al. 2012; Figure 10 and Figure 11a) at about the same time as the quenching of anomalous mantle-derived magma (Days Creek Gabbro chilled margin) in the Tablelands Complex. This was shortly followed by the main group of Bakers Creek Suite gabbroic melts (Days Creek, Barney House and Big Bull) at 305-304 Ma. The earliest, anomalous magmas in the chilled margin have some unusual characteristics, such as a fluid-mobile high Ba component, low Zr, and potentially lower Ti and higher V, as well as a general depletion in trace elements. These can be considered a kind of 'forearc' or ~FAB type component (e.g. DC65) indicating decompression melting of the old, depleted mantle wedge and therefore extension of the overlying forearc basin and accretionary prism (Figure 11b). Evolution to BAB magmatism reflects a combination of a more enriched mantle source and conventional EAT or IAT component. It also suggests continued extension (Figure 11c) related to slab rollback, or perhaps less likely by slab breakoff, and could ultimately be driven by reorganisation of the paleo-Pacific plate. Continued heating and melting of the accretion complex during rifting

generated high-T metamorphic complexes and the earliest S-type granitic melts of the Hillgrove Suite at ~302-300 Ma. These mixed with Bakers Creek Suite gabbroic melts producing a full spectrum from mafic to felsic compositions (Figure 11d). Peak melting of fertile greywackes, probably due to underplating of mafic melts, led to the main flux of S-type granites at ~290 Ma (Figure 11e). While modelling of such processes usually indicates relatively short timescales of ~1 Myr or less for abundant felsic melt production (Annen and Sparks 2002; Solano et al. 2012), the chronology constructed for the NEO implies that melt mobilisation takes significantly longer, perhaps due to rapid stratification of mafic and felsic melts (preventing later mafic melts from ascending) and importantly involving high melt fractions.

**8 Conclusions**

- The Bakers Creek Suite gabbros, associated with the Hillgrove Suite S-type granitoids, record an evolution from an early forearc like component through normal arc-backarc style gabbroic magmatism, to hybrid melts with a wide spectrum of intermediate compositions (continuous with the S-type Hillgrove Suite).

- Capture of a FAB-type component occurred in the early basaltic melts of the Bakers Creek Suite. This earliest intrusive magmatism in the NEO occurred at ~305 Ma and was subsequently replaced by incipient arc-backarc type magma at ~305-304 Ma.

- Rifting, extension of the overlying sedimentary complex, melting of the mantle wedge, and transport of the resulting melts were responsible for the high temperature, low pressure metamorphic complexes in the mid-crust and abundant S-type granitic magmas at depth (~300 Ma and onwards), with peak migration and emplacement of the latter at ~290 Ma.

- Ultimately, formation of a new orogen (New England) from the forearc of the Lachlan Orogen (Tamworth Belt and Tablelands Complex Accretionary prism) occurred by forearc rifting.

*Acknowledgements.* We thank Jenny Zobec (UoN) for XRF analyses, Stephen Eggins (ANU) for LA-ICP-MS analysis, Daniela Rubatto, Trevor Ireland, Peter Holden, and Peter Lanc for assistance with SHRIMP preparation and measurements, and Ian Williams, Richard Arculus, Joerg Hermann, Colleen Bryant, Pengfei Li, Gordon Lister, Bill Collins, and Robin Offler for lengthy discussion about zircon standards, magmatic geochemistry, the NEO and tectonics. This research was partially supported by an Australian National University PhD Research Scholarship to SMcK, who is currently a postdoctoral fellow of the Research Foundation-Flanders (Fonds Wettenschapplijke Onderszoek; FWO).

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



**Figure 1:** (a) New England Orogen and field area (orange) in the Tablelands Complex. (b) Hillgrove and Bakers Creek Suite plutons in the Tablelands Complex; overlying Tertiary basalt omitted for clarity. Sampled Bakers Creek plutons: (1) Mornington; (2) Big Bull; (3) Charon Creek; (4) Days Creek; (5) Camperdown; (6) Bakers Creek; (7) Barney House; (8) Cheyenne; (9) Woodburn; (10) Moona Plains; (11) Apsley River; (12) Eastlake (Hillgrove Suite).




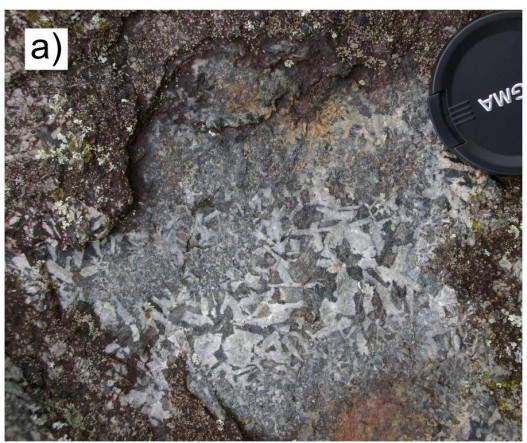

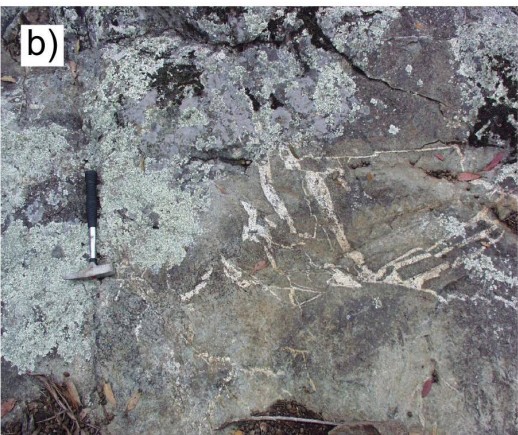

**Figure 2:** Photos of the Days Creek gabbro showing: a) massive gabbro enclosing domain of gabbro pegmatite. Camera lens has a diameter of 5 cm; b) Finer grained dolerite pluton margin hosting felsic veins. Hammer length 30 cm.



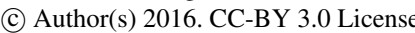

**Figure 3:** Cross-polarised light images of gabbroic rocks; all images field of view 2 mm. a) Chilled, flow-foliated margin of the Days Creek gabbro DC65. b) Ophitic micro-gabbro in Barney House gabbro BH30. c) Phenocrystic plagioclase in Barney House gabbro BH45. d) Relict olivine in massive Days Creek gabbro DC19. e) Granular Days Creek gabbro DC36. f) Poikolitic biotite-diorite associated with Days Creek gabbro DC98.



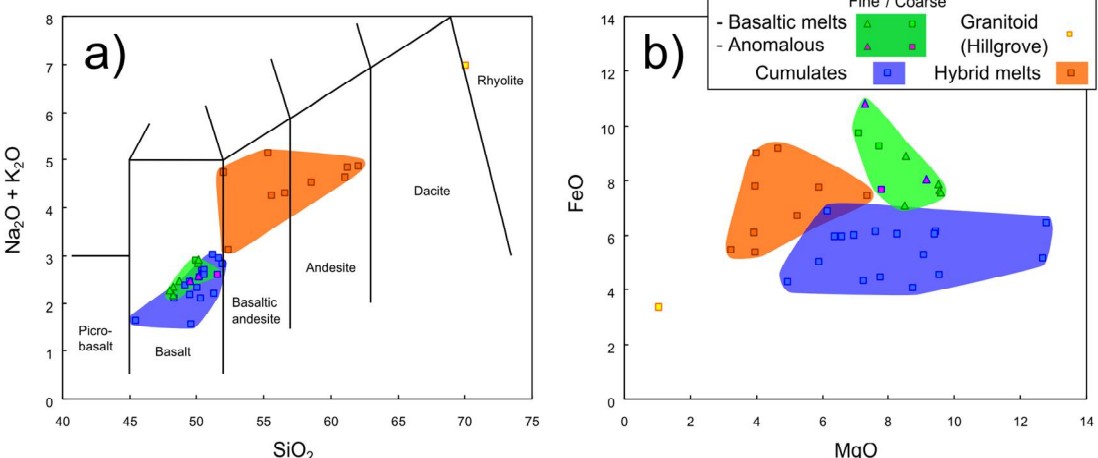

**Figure 4:** Major element chemistry of bulk samples. Finely crystalline, doleritic samples in triangles, coarsely crystalline samples in squares; basaltic and anomalous basaltic samples likely to be melt compositions are in green and pink respectively; cumulate samples in blue; mid-silica samples in orange; yellow indicating a single granitic sample. (a) TAS (total alkalis v. silica) diagram, and (b) MgO v. FeO, with the trend defined by inferred melt compositions (in green field) indicating a 'tholeiitic' association.





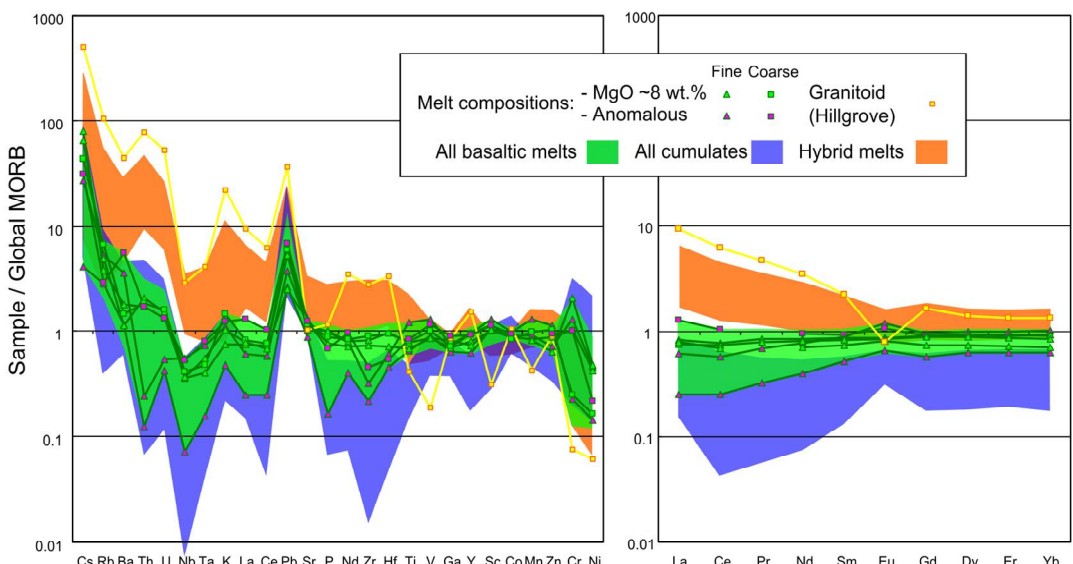

**Figure 5:** Trace element chemistry of bulk samples, normalised to Global MORB (Arevalo and McDonough 2010). Finely and coarsely crystalline gabbros (triangles and squares respectively) that approximate melt compositions with ~8 wt. % MgO in green; anomalous melt compositions in pink; Eastlake monzogranite (Hillgrove Suite) in yellow. Range of compositions for basaltic melts, cumulates, and higher silica hybrid melts are given by green, blue, and orange fields respectively.




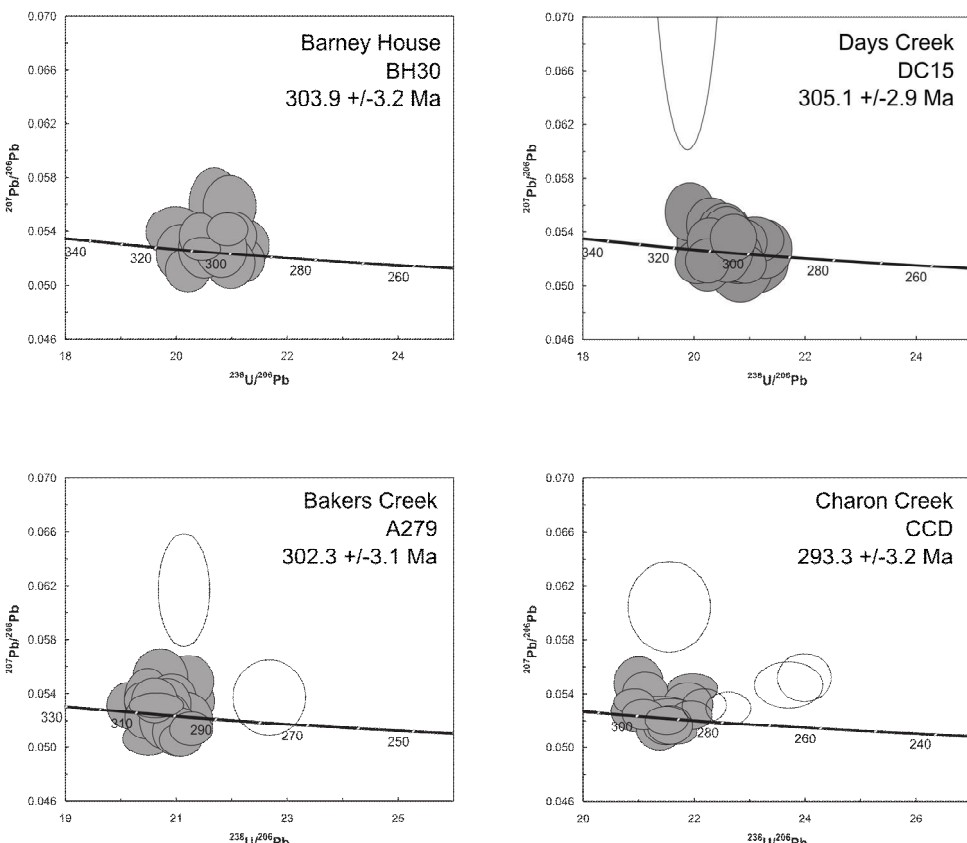

**Figure 6:** Tera-Wasserburg Concordia plots of U-Pb data for Bakers Creek samples. Individual spot error ellipses are 68.3% confidence limits. Unfilled ellipses were not included in weighted mean age calculations; age intervals are 95% confidence and include error on standards.



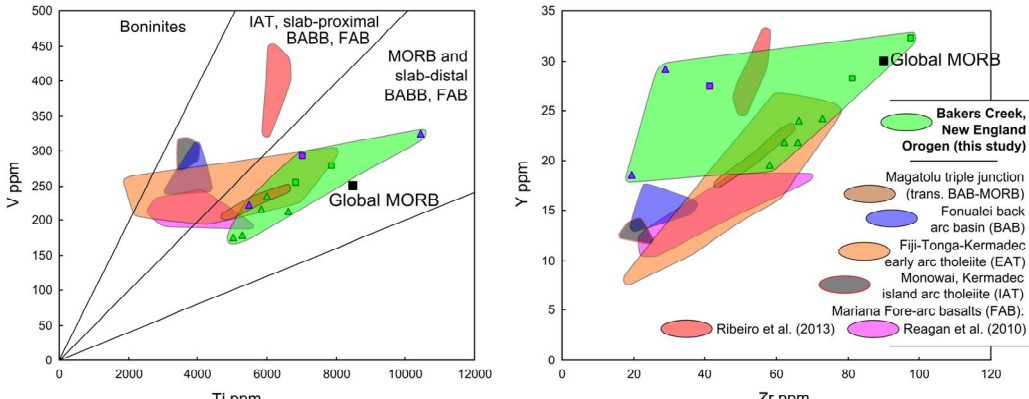

**Figure 7:** Ti-V and Zr-Y systematics of Bakers Creek Suite melt compositions and selected primitive melts from West Pacific arc systems (Keller et al. 2008; Reagan et al. 2010; Timm et al. 2011; Escrig et al. 2012; Ribeiro et al. 2013; Todd et al. 2013; Kemner et al. 2015). Ti-V fields from Shervais (1982) and Pearce (2014).





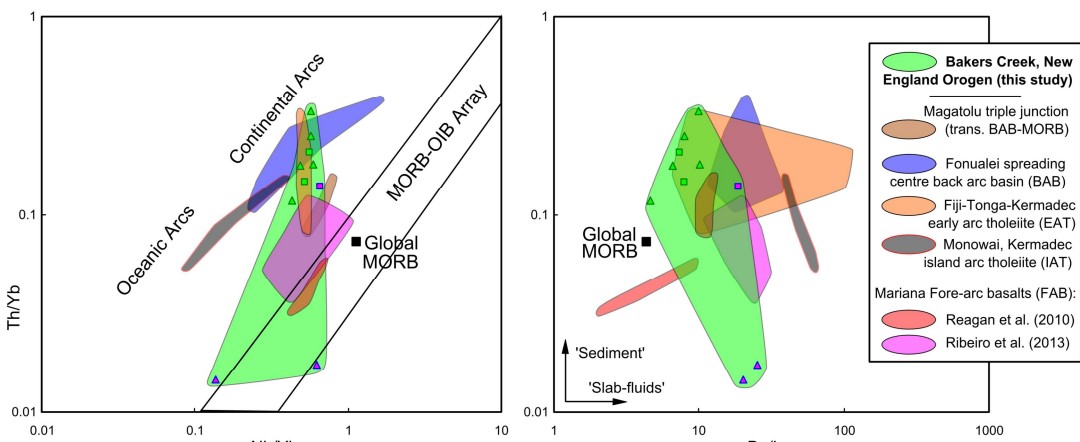

**Figure 8:** Indicators for subduction zone components, using Th/Yb as a proxy for sedimentary melt, versus: (a) Nb/Yb identifying depletion of Nb, after 'classical' subduction zone signatures (Pearce 2014); and (b) Ba/La as a proxy for a fluid-mobile component (Woodhead et al. 2001). Bakers Creek Suite compositions and selected primitive melts from West Pacific arc systems as for Figure 7.




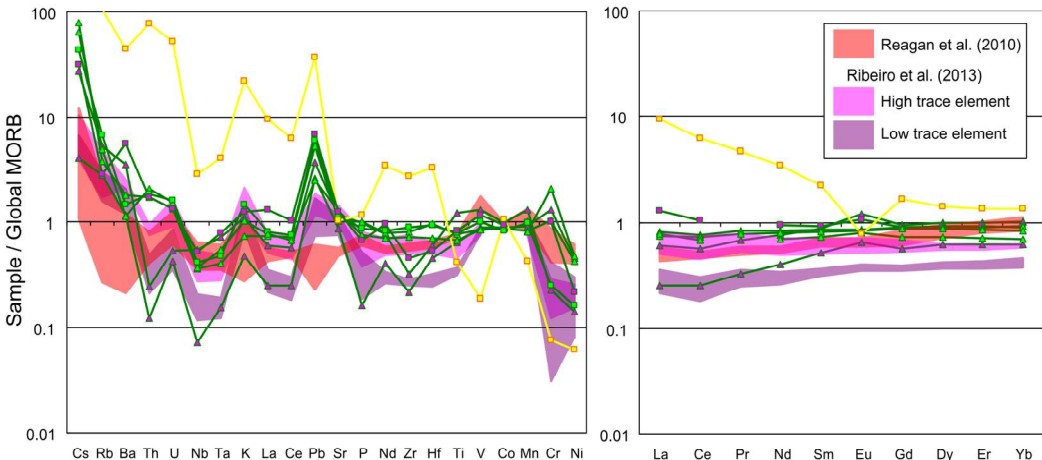

**Figure 9:** Bakers Creek basaltic melt compositions compared to forearc basalts from Reagan et al. (2010) and Ribeiro et al. (2013) for samples with ~6.6-8.6 wt. % MgO.





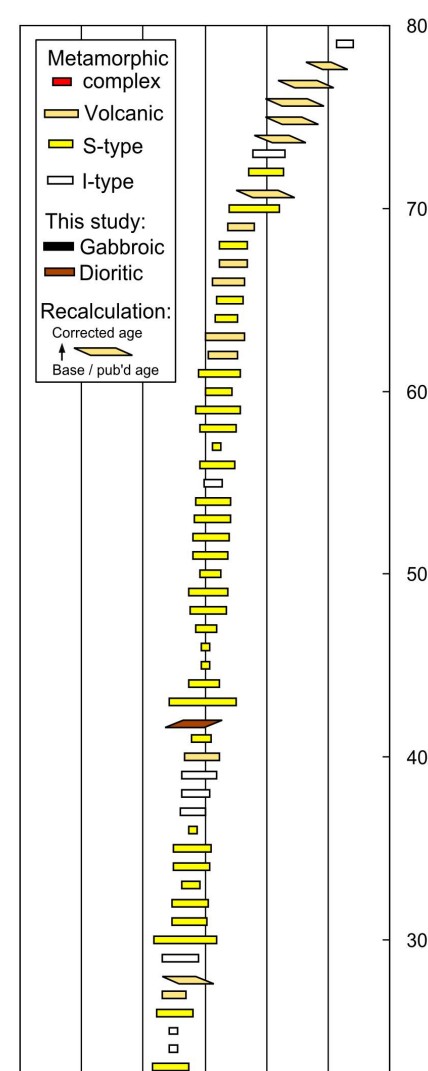





**Figure 10:** Summary of age determinations for latest Carboniferous to early Permian magmatic rocks of the Southern NEO, corresponding to the first, predominantly S-type, granitic magmatism in the orogen. Sources of data in Supplement 4.

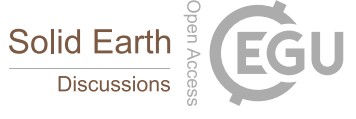

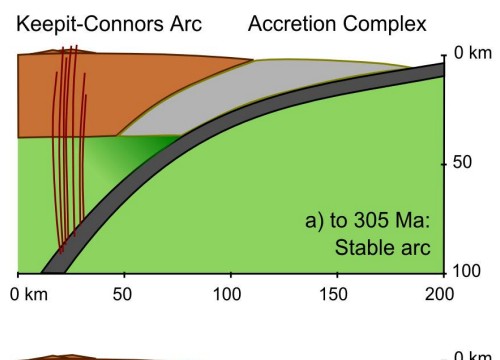

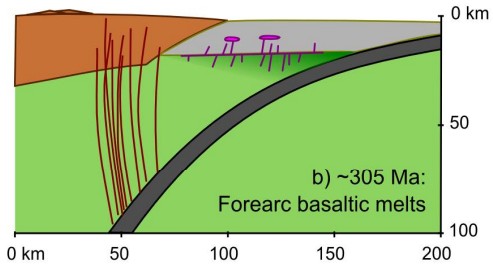

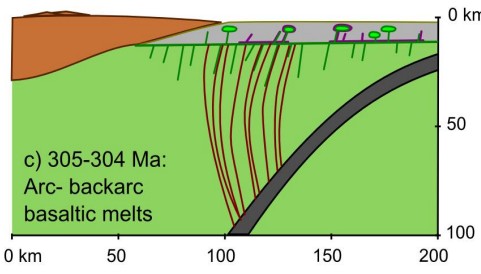

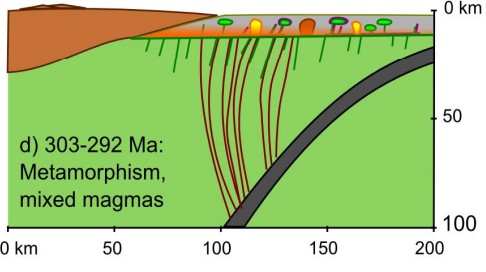

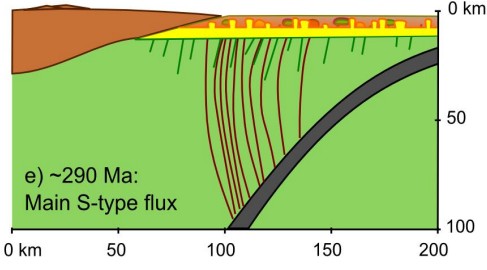



**Figure 11:** Chronology of early NEO magmatism: a) Lachlan Orogen subduction zone (Keepit-Connors arc) at ~310-305 Ma; b) Early extension and production of forearc basalt (FAB) type melts, emplaced as rare, anomalous chilled margins of the Bakers Creek Suite at ~305 Ma; c) Continued extension at 305-304 Ma inducing mantle melting and production of main Bakers Creek Suite gabbros with backarc (BAB) to arc-like (EAT, IAT) affinities; d) Peak metamorphism in metamorphic complexes (Tia and Wongwibinda) and diverse magmatism including mafic and felsic components (Bakers Creek and Hillgrove Suites) at 303-292 Ma; e) Main flux of Hillgrove and Bundarra S-type granites at ~290 Ma.