# Peer review of "First magmatism in the New England Orogen, Australia: Forearc and arc-backarc components in the Bakers Creek Suite gabbros"

_Solid Earth, 2016_

## Short Comment (SC1) · 2 Sep 2016

This is an interesting paper. I am wondering where is the early Permian arc preserved. Is it possible to preserve in the Gympie terrane? You may check the recent paper in tectonics:

Li, P., Rosenbaum, G., Yang, J.-H., and Hoy, D., 2015, Australian-derived detrital zircons in the Permian-Triassic Gympie terrane (eastern Australia): evidence for an autochthonous origin: Tectonics, v. 34, p. 2015TC003829, doi: 10.1002/2015TC003829.

---

## Short Comment (SC2) · 11 Sep 2016

Dear Sean,

Thanks for giving me a chance to read your interesting paper. I'd like to simply mention two things:

1. I am not fully aware of the relationship between the Bakers Creek Suite and Hill-grove Supersuite. They are spatially associated, but was it demonstrated that they are cognate or related petrogenetically? I know that you compared the whole rock geo-chemistry of Bakers Creek and Hillgrove in NEO 2010 conference proceeding (I may not remember clearly and cannot find that proceeding paper), which you may be able

to mention in this paper?

2. Glad to see that you made criteria of Th/U to distinguish zircon ages for the granite crystalline from those of inherited. With no O isotopic data, that is probably the best way to exclude "possible" inherited zircon ages. But as the zircon Th/U ratio of 0.3 is not an absolute reference, it is also very useful to check if individual dating spot is on the clear magmatic rim (Jeon et al. [2012, EPSL] observed that all measured inherited cores and texturally discordant cores have thick overgrowth magmatic rim).

Great work,

Heejin Jeon

───────────────────────

---

## Short Comment (SC3) · 29 Sep 2016

It is generally accepted that the New England Orogenic phase began with the end of the Kanimblan compression in the Lachlan Orogen and accretion of the Gamilaroi/Calliope island arcs. The Carboniferous arc and associated forearc and accretionary complex are part of the New England Orogen. (e.g. Schiebner 1997 - The Geology of NSW)
* * *

---

## Referee Comment (RC1) · Anonymous Referee #1 · 19 Oct 2016

This paper presents new petrological, geochemical and geochronological data, which are helpful for understanding the late Paleozoic tectonic evolution along the eastern Australian margin. The data quality is good, and the interpretation is reasonable. Here I provide some suggestion for authors to further improve the manuscript.

(1) A schematic diagram is required to show the major tectonic element of the NEO. This is particularly important for the regional geology section. Otherwise, it is difficult for readers to follow when seeing some terms (the Tamworth Belt, the Tablelands Complex and so on). Authors may be able to refer to Fig 1 in Li et al (2015) or Fig 1 in Glen and Roberts (2012).

(2) Section 7.3 for tectonic implication is kind of weak. Actually, there are a large

number of structural, metamorphic, sedimentary, magmatic data to support authors' interpretation. The discussion will be significantly strengthened if these data can be incorporated. Authors may also think about discussing a bit how the Permian tectonic units in New Zealand and New Caledonia, and the Permian Gympie terrane are linked with the tectonic transition mentioned in the manuscript.

(3) Section 7.2 refers to a large number of intrusion names. These names should be somehow demonstrated in a figure. Otherwise, it will be difficult for readers to find out where are these rocks.

Additional minor comments:

Page 2_Line 27: Using "eastward" to replace "outboard"

Page 3_line 3: Actually, only that part of the Hillgrove Suite close to the shear/fault zone is foliated. As far as I know, some parts of the Hillgrove Suite are non-foliated.

Page 10_Line 18: Li et al. (2014) also dated the basalt of the Alum Mt Volcanics, which yielded an eruption age at around 272 Ma. This age is similar as the SHRIMP zircon age from the felsic part. Such information should be provided.

Page 10_Line 23: This statement for the Lachlan Orogen is confused. The orogenesis for the New England Orogen had already initiated in the Late Devonian.

Page 10_Line 34: melting of the Tablelands accretionary complex

References: Glen, R.A., Roberts, J., 2012. Formation of oroclines in the New England Orogen, Eastern Australia. J. Virtual Explor. 43, Paper 3. Li, P., Rosenbaum, G., Vasconcelos, P., 2014. Chronological constraints on the Permian geodynamic evolution of eastern Australia. Tectonophysics 617, 20-30. Li, P., Rosenbaum, G., Yang, J.-H., Hoy, D., 2015. Australian-derived detrital zircons in the Permian-Triassic Gympie terrane (eastern Australia): evidence for an autochthonous origin. Tectonics 34, 2015TC003829, doi: 10.1002/2015TC003829.

---

## Referee Comment (RC2) · Anonymous Referee #2 · 21 Nov 2016

This study presents a new set of geochemical data and U-Pb ages on zircons from the Bakers Creek suite Gabbros. Theses new data are are used to constrain the tectonic settings of the first magmatism of the New England Orogen. As a non-specialist of geochronology, I have no comments on the zircon chronology work and I leave its evaluation to specialists. I provide here a review on the work related to the geochemistry of major and trace elements.

Major comments:

First, I regret to say that the analytical section suffers from the lack of results on geological reference materials and information on the limits of detection, quantification and LA-ICP-MS settings. Second, I was not convinced by the use of the major and trace

element data sets proposed by the authors. Bellow, I report some examples illustrating (1) that the data presentation suffers from the lack of clarity (definition of sample grouping, sample selection, etc.), and (2) that the interpretations are not supported by the use of trace and major elements.

1. Analytical session.

Page 3 line 27: "some trace elements": which ones did you analyzed by XRF and how does these data compared to LA-ICP-MS data? How did the authors measure the L.O.I.? Page 3 line 32: please provide the ICP-MS and laser settings (laser energy, laser shots frequency, spot or raster ablation mode, etc.). Which NIST glasses is used for the calibration and what are the reference values used for this NIST? Please provide also the detection limits. What is the purity of the lithium borate flux used for the fusion? Could this be a concern for the sample characterized by very low trace element contents (e.g. Th< 0.05 ppm)? Finally, did the authors analyze any of the BIR-1g, BHVO-2g, etc. reference materials to certified their analytical protocol?

2. Data presentation

- In figures 4 and 5, the data are sub-divided into basaltic melts (which corresponds in reality to finely and coarse crystalline gabbros), cumulate rocks and hybrid melts. Because the term "Hybrid melts" is not mentioned at all in the discussion or in the data table, the reader has no clues about the nature and origin of this group of sample. To which sample these hybrids melts corresponds to? What does the term "hybrid" stand for? - The authors mentioned also "anomalous samples" in figure 4 and 5. How do they define the anomalous character of these samples? My guess is that theses samples correspond to those analyses with extremely low Th contents (i.e. < 0.05 ppm), which could potentially be close or even below the limit of quantification. This observation echo's my comment on the analytical session.

These two examples show that it is not very easy (or very time consuming) to understand and follow the links between between the figures, the text, and data tables.
2- data interpretation

Because of the lack of clarity (linearity) in the data presentation, I was not really able to evaluate properly the geochemical interpretation of major and trace elements data. Nonetheless, I address bellow few major comments for the authors.

- The authors use the geochemistry of Large Ion Lithophile Elements (LILE) to demonstrate the arc-sub-arc settings of Baker Creek suite gabbros (page 9 line 4-9 and figures 5a and 9a). Given the age of these samples ($\sim$ 300Ma), the authors should first demonstrate that the LILE abundances of these samples have not been modified by alteration. - As far as I understand, the Th/Yb ratio is the only non-LILE trace element ratio that suggests a sub-arc setting for Bakers Creek gabbros. This result should also be confirmed by the use of other trace element ratios such as Nb/La, Nb/Ta and Th/La. Note that I do not see any evidence for a sub-arc setting from the trace element ratios involving Ti, Zr, Y and V. - The MORB reference should not be restricted to one point. Please report the MORB field instead of a single point. Baker Creek gabbros might certainly overlap with MORB data in figures 7. - The sub-arc setting is discussed only on the basis of 10 analyses of backers Creek gabbros. Five of theses analyses display an "anomalous signature" or correspond to coarsely crystalline gabbros that may not be representative of melt compositions. How representative are the geochemical results of Bakers Creek gabbros in this context? What is the story of the "Hybrid melts" and "cumulate" rocks? Is it compatible with the one from Bakers Creek gabbros? - The role of crustal contamination is also not discussed in this paper. Is it possible for example that the high Th/Yb ratios measured in Baker creek sample (Fig 8a) could result from crustal assimilation?

Minor comments

Page 6 line 8: the figure shows only FeO vs. MgO. Page 6 line 9: repot the MORB and BABB fields in Fig 4.b. Page 6 line 24: It is not possible to see clearly theses samples in Fig. 5. There are 5 different symbols and 3 different colors. Page 7 line

1: replace "peaks" by "anomalies" Page 8 line 10: "Magmatic differentiation occurred before or during emplacement of magmas at depth in the mantle wedge" Why would differentiation occur within the mantle wedge? Page 8 line 15: I found the uses of the term "melt" abusive for the chemical composition of plutonic rocks. I think that this statement needs to be discussed and argued in the text. Page 9 line 18: please specify the nature of the components. Data table: The data table could benefit from the addition of petrography information (grain size, cumulate, chilled margin, etc.).

---

## Author Comment (AC1) · 27 Feb 2017

The authors thank the two anonymous reviewers and the three formal commentators on our submitted manuscript. We have used these constructive points to formulate a revised version of our manuscript that we herewith re-submit to Solid Earth.

The comments and reviews dealing with:

SC1: Location of the volcanic arc during the earliest Permian SC2: U-Pb zircon rejection criteria SC3: The age of major regional units and our terminology for the New England Orogen RC1: General tectonics of the New England Orogen RC2: Geochemical analysis of magmatic rocks and tectonic implications following therefrom

[Figure]

...have been addressed in a point-by-point reply (2017-01-09 RESPONSE McKIBBIN et al Bakers Creek gabbros.pdf), and changes relating to these comments and reviews are documented in a track-changes version of the manuscript (2017-02-25 McKIBBIN et al Bakers Creek gabbros REVISED MANUSCRIPT.pdf).

All documents, figures, and supplementary materials are contained in the zip file below.

Once more we thank the reviewers, commentators and editorial staff for their assistance with our manuscript.

Regards,

Seann McKibbin and co-authors

Please also note the supplement to this comment:
http://www.solid-earth-discuss.net/se-2016-123/se-2016-123-AC1-supplement.zip